# Spatiotemporal Distribution of Hand, Foot, and Mouth Disease in Guangdong Province, China and Potential Predictors, 2009–2012

**DOI:** 10.3390/ijerph16071191

**Published:** 2019-04-03

**Authors:** Yijing Wang, Yingsi Lai, Zhicheng Du, Wangjian Zhang, Chenyang Feng, Ruixue Li, Yuantao Hao

**Affiliations:** 1Department of Medical Statistic and Epidemiology, School of Public Health, Sun Yat-sen University, 74 Zhong Shan 2nd Road, Guangzhou 510080, China; wangyj77@mail2.sysu.edu.cn (Y.W.); laiys3@mail.sysu.edu.cn (Y.L.); duzhch3@mail2.sysu.edu.cn (Z.D.); fengchy3@mail2.sysu.edu.cn (C.F.); lirx6@mail2.sysu.edu.cn (R.L.); 2Sun Yat-sen Global Health Institute, Sun Yat-sen University, 135 Xin Gang Xi Road, Guangzhou 510275, China; 3Key Laboratory of Tropical Diseases and Control of the Ministry of Education, Guangzhou 510080, China; 4Department of Environmental Health Sciences, School of Public Health, University at Albany, State University of New York, Rensselaer, NY 12144, USA; wzhang27@albany.edu

**Keywords:** hand, foot, and mouth disease, Bayesian spatiotemporal models, spatiotemporal analysis, spatiotemporal interaction, potential predictors

## Abstract

*Background*: Hand, foot, and mouth disease (HFMD) is a common infectious disease among children. Guangdong Province is one of the most severely affected provinces in south China. This study aims to identify the spatiotemporal distribution characteristics and potential predictors of HFMD in Guangdong Province and provide a theoretical basis for the disease control and prevention. *Methods*: Case-based HFMD surveillance data from 2009 to 2012 was obtained from the China Center for Disease Control and Prevention (China CDC). The Bayesian spatiotemporal model was used to evaluate the spatiotemporal variations of HFMD and identify the potential association with meteorological and socioeconomic factors. *Results*: Spatially, areas with higher relative risk (*RR*) of HFMD tended to be clustered around the Pearl River Delta region (the mid-east of the province). Temporally, we observed that the risk of HFMD peaked from April to July and October to December each year and detected an upward trend between 2009 and 2012. There was positive nonlinear enhancement between spatial and temporal effects, and the distribution of relative risk in space was not fixed, which had an irregular fluctuating trend in each month. The risk of HFMD was significantly associated with monthly average relative humidity (*RR*: 1.015, 95% *CI*: 1.006–1.024), monthly average temperature (*RR*: 1.045, 95% *CI*: 1.021–1.069), and monthly average rainfall (*RR*: 1.004, 95% *CI*: 1.001–1.008), but not significantly associated with average GDP. *Conclusions*: The risk of HFMD in Guangdong showed significant spatiotemporal heterogeneity. There was spatiotemporal interaction in the relative risk of HFMD. Adding a spatiotemporal interaction term could well explain the change of spatial effect with time, thus increasing the goodness of fit of the model. Meteorological factors, such as monthly average relative humidity, monthly average temperature, and monthly average rainfall, might be the driving factors of HFMD.

## 1. Introduction

Hand, foot, and mouth disease (HFMD), primarily caused by Enterovirus 71 (EV71) and Coxsackievirus A16 (Cox A16) [1], is a common infectious disease among children [2]. HFMD is typically characterized by fever, skin eruptions on hands, feet, buttocks, and vesicles/ulcers in the mouth. It can be transmitted through various routes, including direct contact with fluid from blisters, and inhaling the virus through the respiratory tract [3]. HFMD epidemics have been reported worldwide, such as in Singapore [4], Vietnam [5], Hong Kong [6], and Japan [7]. China is one of the most severely affected countries. The overview of the national notifiable infectious disease epidemic in China showed that the HFMD incidence in 2017 ranked the first, followed by hepatitis and tuberculosis [8]. Through 2008 to 2012, at least 6.5 million pediatric cases of HFMD were recorded, of which more than 2000 died [9]. The situation seems worse in Guangdong Province where the provincial incidence was 3–4 times higher than the national average [10,11]. Although the EV 71 vaccine has been available since 2016, the HFMD epidemic remains urgent.

Previous studies have demonstrated geographical and temporal variations of HFMD across China. In China, the incidence of HFMD varies in different provinces. In 2013, the incidence was reported to be 67.90/100,000 in Sichuan Province [12], while it was 100/100,000 in Chongqing city [3]. Even in the same province, the incidence rate also varied across counties [13,14]. Different seasonality of HFMD appears. In Hong Kong, a seasonal peak was detected in the warmer months (May–July), along with a smaller winter peak (October–December) [15]. Whereas in Beijing, the seasonal peak occurred in May to July every year [13]. The peaks had slight differences among different regions.

Prior studies suggested that meteorological factors such as temperature and relative humidity might be important predictors for HFMD, with higher temperature or higher relative humidity associated with higher risk of HFMD [12,13,16]. In addition, other meteorological factors, such as rainfall, wind speed, and sunshine duration, also appeared to influence HFMD transmission [13,17]. Socioeconomic variables may also be associated with HFMD incidence. A study in Sichuan showed that HFMD incidence was higher in urban areas compared to rural areas and per capita gross domestic product (GDP) was a risk factor associated with HFMD incidence [12]. Although the potential associations between the risk of HFMD and meteorological/socioeconomic factors were reported previously, the findings might be not applicable to Guangdong due to the spatial differential.

It is necessary to conduct a comprehensive analysis of HFMD, including the effects of space-time and potential predictors. However, most of the prior studies only focused on the potential predictors or the cluster analysis [3,18,19,20]. Few studies can be conducted in the perspective of spatiotemporal interaction, taking into account the potential predictors, as well as the spatiotemporal dynamic change of HFMD.

The Bayesian spatiotemporal model we used includes the spatiotemporal effects to the generalized additive model to identify spatiotemporal variations and the effects of potential predictors at the same time [21]. This approach can also control uncertainties resulted from residual confounding, i.e., confounding that is not included in the model, by incorporating spatial and temporal random terms.

Therefore, this study aims to analyze the spatiotemporal distribution and potential predictors of HFMD in Guangdong Province after controlling multiple factors, including spatial effect, temporal effect, spatiotemporal interaction effect, and potential predictors, and provide information and theoretical basis for the disease control and prevention.

## 2. Materials and Methods

### 2.1. Study Area

Guangdong Province, a large coastal province in south China, has an area of 179,800 km^2^ and a population of 106 million (from the 2013 China Statistical Yearbook). There were 21 administrative districts, comprising of 123 counties in Guangdong Province. We used the county level division as the geographical unit for this spatiotemporal analysis because the county administrative level was often used for governmental statistics.

According to the characteristics of the natural landscape and economic development, Guangdong Province can be divided into two or four parts: the Pearl River Delta region (including the capital city Guangzhou) and the non-Pearl River Delta region; or the Pearl River Delta region, eastern Guangdong region, western Guangdong region, and northern Guangdong, as shown in Figure 1. The Pearl River Delta region has a much higher level of socioeconomic development, accounting for 80% GDP of the whole province with less than 50% population [10].

### 2.2. Surveillance Data of HFMD

Case-based HFMD surveillance data from 2009 to 2012 was obtained from the China Center for Disease Control and Prevention (China CDC). It was confidential data; raw data cannot be published due to state law and ethics. The clinical criteria for diagnosis of HFMD cases was provided in a guidebook published by the Ministry of Health of China in 2008 [22], in which patients were defined as HFMD with the occurrence of the following symptoms: fever, papules and herpetic lesions on the hands or feet, rashes on the buttocks or knees, inflammatory flushing around the rashes and little fluid in the blisters, sparse herpetic lesions on oral mucosa.

### 2.3. Potential Predictors Data

Monthly average relative humidity, the monthly average temperature, monthly average rainfall, monthly average wind speed, and monthly sunshine duration data were obtained from the China Meteorological Data Sharing Service System (http://data.cma.cn/). The monthly county-level meteorological variables were estimated using ordinary spatial kriging methods based on 26 meteorological surveillance stations within Guangdong Province. The 26 meteorological surveillance stations are mapped in Figure 2. In addition, we obtained population and socioeconomic data, i.e., average GDP, from the statistical yearbook.

### 2.4. Statistical Methods

#### 2.4.1. Bayesian Spatiotemporal Model

The Bayesian spatiotemporal model was used to analyze the relative risk (*RR*) of HFMD from 2009 to 2012. The form of the model was specified as the following:(1)$$\begin{array}{l} Y_{ij}∼Poisson(\lambda_{ij})\\ E(Y_{ij})=\lambda_{ij}=e_{ij}\times \theta_{ij}\\ \log(\theta_{ij})=b_{0}+\sum_{p}\beta_{p}X_{pij}+\mu_{i}+\upsilon_{i}+\gamma_{j}+\varphi_{j}+\delta_{ij} \end{array}$$
where $Y_{ij}$ was the number of reported HFMD cases in region i, month j (j = 2009-01, …, 2012-12). The model assumed that $Y_{ij}$ was subject to a Poisson distribution with a mean $\lambda_{ij}=e_{ij}\times \theta_{ij}$, where $e_{ij}$ represented the expected number of HFMD cases, which was calculated as the product of overall incidence of the province and the population for each county during the study period, and $\theta_{ij}$ was the *RR*. $b_{0}$ was the intercept, $X_{pij}$ was the $p_{th}$ meteorological or socioeconomic variable in region i, month j, $\beta_{p}$ was the regression coefficient. We calculated the variance inflation factor (VIF) for all candidate variables to assess the multicollinearity; variables with VIF less than 5 were selected for inclusion in the model [23]. The Besag, York, and Mollie (BYM) model was used to model the common spatial component [24], which consisted of two components: spatially structured random effect $\mu_{i}$ and spatially unstructured random effect $\upsilon_{i}$. $\mu_{i}$ was commonly assumed to follow a conditional autoregressive prior structure (CAR) in infectious diseases spatial epidemiology [25,26,27], which showed that county i had a similar pattern of disease incidence with the adjacent counties. $\upsilon_{i}$ was assumed to be a normal distribution, representing that county i had an independent pattern of disease incidence from the adjacent counties. If a disease has a strong spatial autocorrelation, the spatially structured random effect $\mu_{i}$ will have more contribution explaining the spatial variance of the disease risk; otherwise, the component of the spatially unstructured random effect $\upsilon_{i}$ will have more explanatory power. The temporal component was also consisted of two components: monthly structured random effect $\gamma_{j}$ and monthly unstructured random effect $\varphi_{j}$. $\gamma_{j}$ and $\varphi_{j}$ were assumed to follow a 1 order random walk and a normal distribution, respectively [21,28,29]. $\delta_{ij}$ was the spatiotemporal interaction term. The structure matrix can be written as the Kronecker product of $R_{\delta }=R_{\upsilon }⊗R_{\varphi }$, which assumed that the spatially unstructured random effect $\upsilon_{i}$ and the monthly unstructured random effect $\varphi_{j}$ interact with each other [21].

We developed five different models. Model 1 only included potential predictors. Model 2 and model 3 added a spatial component and a temporal component to model 1, respectively. Model 4 was the combination of model 2 and model 3. Finally, model 5 included all the components, including potential predictors, spatial, temporal, and spatiotemporal interaction components.

The models were fitted using the integrated nested Laplace approximation (INLA) method incorporated in the R-INLA package (R version 3.4.3., R Core Team, Vienna, Austria). The model with the smallest deviance information criterion (DIC) was considered the optimal and used for our study.

#### 2.4.2. Geographical Detectors

In this study, the *q*-statistic was used to quantify the spatiotemporal heterogeneity of HFMD and detect the interaction relationship between spatial effect and temporal effect [30,31]. The form of the *q*-statistic was specified as the following:(2)$$q=1-\frac{1}{N\sigma^{2}}\sum_{h=1}^{L}N_{h}\sigma_{h}^{2}$$
where *q* denotes the level of spatiotemporal heterogeneity; the value of the statistic is required to be within [0, 1]. If the value approaches 1, it indicates a strong heterogeneity and if the value approaches 0, it indicates a random distribution. N was the number of all units, which could be divided into L stratums. Stratum h was composed of $N_{h}$ units. $\sigma^{2}$ and $\sigma_{h}^{2}$ were the variance over all the units and within stratum h
(h=1,…,L), respectively.

We can divide the units of observation into two stratums, space $L_{S}$ ($L_{S}$ = 1, …, 21) and time $L_{T}$ ($L_{T}$ = 1, …, 48). If $q(L_{S}\cap L_{T})>q(L_{S})+q(L_{T})$, there is a nonlinear enhancement of space and time, if $q(L_{S}\cap L_{T})<Min(q(L_{S}),q(L_{T}))$, there is a nonlinear weakening of them [31].

## 3. Results

### 3.1. Descriptive Statistics

Overall, 911,640 HFMD cases were reported in Guangdong Province from 2009 to 2012, and the annual average incidences from 2009 to 2012 were 10.36/10,000, 24.52/10,000, 28.21/10,000, and 35.26/10,000, respectively. Spatial distribution of cumulative incidence of HFMD at county level during the study period is shown in Figure 1. Counties with a higher HFMD incidence were clustered in the Pearl River Delta region. A descriptive summary of the meteorological and socioeconomic variables is shown in Table 1.

### 3.2. Model Selection

Table 2 shows the results of the multicollinearity analysis. The VIF values of all variables were less than 5, so it could be considered that there was no multicollinearity between the variables, and all variables were included in the model.

Table 3 summarizes the DIC values for the five types of models. Model 5 was selected as the final model as it had the lowest DIC value. Further results were all based on model 5.

### 3.3. Spatial Distribution

The estimated *RR* values of the spatial effect ($\mu_{i}+\upsilon_{i}$) are mapped in Figure 3. The spatial distribution of HFMD risk exhibited explicit spatial heterogeneity, as the *q*-statistic was 0.669 (*p* < 0.001). The *RR* values were relatively high in the Pearl River Delta region of Guangdong Province. This finding implied that the counties in the Pearl River Delta region had a relatively higher HFMD risk.

If a county had a stable higher or lower risk compared with the overall level in the region, it was classified as a hot spot or a cold spot, respectively [32]. This classification can make the spatial distribution of *RR* more intuitive. Hot and cold spots can be calculated according to the posterior probability of spatial effect—if the posterior probability $P(exp(\mu_{i}+\upsilon_{i})>1|data)>0.8$, then the county i was classified as a hot spot. Similarly, if the posterior probability <0.2, then the county i was classified as a cold spot. Figure 3 shows the distribution of hot and cold spots of HFMD in Guangdong Province. There were 53 counties that accounted for 43% of all counties, and they were classified as hot spots. They were mainly distributed in the Pearl River Delta region. The cold spot regions were predominantly clustered in the east-west coast and northern mountainous counties (57 counties), accounting for 46% of all counties.

### 3.4. Temporal Distribution

The yearly and monthly numbers of cases are shown in Figure 4a,b, respectively. It was noteworthy that the incidence of HFMD had an increasing trend. Besides, seasonal peaks were detected around May and October.

The *RR* of monthly effect ($\gamma_{j}+\varphi_{j}$) is shown in Figure 4c. We detected an upward trend in *RR* between 2009 and 2012. The *RR* of monthly effect presented significant seasonality. Semiannual peaks were observed during the study period. The peaks occurred from April to July, and October to December in each year, with the first peak higher than the second.

### 3.5. Spatiotemporal Interaction Patterns

The results of interaction detection showed that there was positive nonlinear enhancement between spatial and temporal effects. $q(L_{S}\cap L_{T})=0.759$ was greater than the sum of $q(L_{S})=0.373$ and $q(L_{T})=0.132$. However, in the previous studies, spatial effect was assumed to be fixed and did not change over time [13,16], i.e., spatial and temporal effects were independent of each other. Actually, the distribution of *RR* in space was not the same in each month, and there were certain changes. The spatiotemporal interaction component $\delta_{ij}$ could explain the differences in the time trend of *RR* for different areas. The results are listed in Figure 5. From the figure, we can see the variations of spatial effect during each month. The results of heterogeneity detection are listed in Table 4. There were 33 months that had no spatial heterogeneity in the distribution of the changed *RR*. Therefore, most of the variations of spatial effect during each month were irregular.

### 3.6. Potential Predictors

The estimated results of regression coefficients of meteorological variables and socioeconomic variables ($\beta_{p}$) are shown in Table 5. There was a positive association between meteorological variables and relative risk. The increasing of one unit in the monthly average relative humidity, monthly average temperature, and monthly average rainfall was associated with the increase of 1.5% (95% CI: 0.6%–2.4%), 4.5% (95% CI: 2.1%–6.9%), and 0.4% (95% CI: 0.1%–0.8%) in the *RR* of HFMD, respectively. Among these variables, the monthly average temperature had the greatest impact on the HFMD. However, monthly sunshine duration, monthly average wind speed, and average GDP were not significantly associated with the risk of HFMD in this study.

Figure 6 shows the county-level estimated *RR* values of associated potential predictors, which were the product terms of the regression coefficients of potential predictors and the average of the standardization values of potential predictors in each region. They represented the *RR* of HFMD caused by potential predictors to each region. The western coast of Guangdong Province had a higher average relative humidity level, resulting in a higher *RR*, as shown in Figure 6a. The southern regions with higher temperatures and the southwestern regions with higher rainfall also had greater *RR*, as shown in Figure 6b,c.

## 4. Discussion

The present study explored the spatiotemporal distribution of HFMD and its associations with potential predictors. The results revealed that the counties with high relative risk were mainly in the Pearl River Delta, which is the economic center of Guangdong. This finding was consistent with previous studies [3,17]. A study of Beijing–Tianjin–Hebei found that high risk regions were mainly located in large cities, such as Beijing, Tianjin, Shijiazhuang, and their neighboring regions [17]. In addition, a study in Chongqing found that cluster centers were in nine main urban districts [3].

During the four-year study period from 2009 to 2012, an increasing trend of relative risk was detected. HFMD showed semiannual seasonality in Guangdong, which was also commonly observed in other southern provinces and regions in China, such as Jiangsu (peaks period: April to June, and November to December) [33], Chongqing (April to July, and October to December) [3], and Hong Kong (May to July, and October to December) [15]. Meanwhile, a single peak has been shown to appear in northern China, such as Beijing (May to July) [13] and Shandong Province (April and August) [34]. Different climates and living habits can be potential reasons for the different peak months. For instance, the climate is warm and humid in southern China, while cold and dry in the north.

Temperature and relative humidity were associated with the relative risk of HFMD, with increased relative risk of the disease following higher temperature and relative humidity. This was consistent with the previous findings in Guangdong [10,35], Beijing [13], and Hong Kong [36]. Average rainfall was also an important influence on HFMD transmission. It was generally consistent with other studies conducted in Beijing and Henan [13,27]. According to previous studies, there are two potential ways that meteorological factors influence HFMD—one is by affecting the survival and reproductive capacity of enteroviruses [17], and another is by altering patterns of human behavior [37]. For example, Huang et al. found that in hot and humid summers, outdoor activities were reduced, and people tended to spend more time indoors in air-conditioned houses, thereby providing more opportunities for contact with each other [35].

The confounders were controlled by adding the spatial and temporal components, which had both effects on the number of reported HFMD cases and meteorological or socioeconomic variables. Spatiotemporal effects represented the residual caused by latent variables that were not included in the model, such as medical conditions and the capacity for HFMD prevention and control in different counties and different periods, which are easier to control than meteorological factors. Thus, areas and months with high relative risk should be paid more attention through the strategies including strengthening the education and supervision of kindergartens.

Major prior studies used traditional models such as spatial autocorrelation and spatiotemporal scan statistics [38]. Spatial autocorrelation cannot detect trends in the time, but just performs spatial analysis. Spatiotemporal scan statistics is a simple and easy method, and it has been widely used in public health. However, covariates cannot be included in the model and it is not suitable for a risk region with an irregular shape. A Bayesian spatiotemporal model is more flexible in modelling spatiotemporal components than the above models with some limitations.

This was a comprehensive study of HFMD in terms of spatiotemporal effects and meteorological variables. We incorporated spatiotemporal effects in the regression that controlled uncertainties resulting from residual confounding, which made the estimation of the impact of meteorological variables more accurate. In addition, the hot spots and high-risk time of HFMD were identified. Through the detection of spatiotemporal interaction, we found that the spatial effect of HFMD had positive nonlinear enhancement with the temporal effect. After adding the spatiotemporal interaction component to the model, the goodness of fit of the model was significantly increased. Compared with the model without the interaction term (model 4), the DIC value was reduced by 77.8% to 45,036.5. Therefore, adding the interaction component can improve the accuracy of the model—this has not been considered in many previous related studies.

The present study has some limitations. First, we used the annual average GDP for each county as the monthly data. Second, there were only 26 meteorological surveillance stations in Guangdong Province; ordinary kriging interpolation results might not cover all variations of the meteorological variables at the county level. However, this was more accurate than using the same meteorological data to all counties in each city. Third, we estimated spatiotemporal variations in HFMD at the scale of counties and months; we did not include the factors at an individual and pathogenic level, such as personal hygiene, educational background, incomes of the children’s parents, living conditions, and composition of major pathogens. Further studies should consider the potential impacts of these factors.

## 5. Conclusions

The risk of HFMD in Guangdong showed significant spatial heterogeneity, with higher relative risk counties mainly gathered in the Pearl River Delta region. The relative risk of HFMD exhibited an increasing trend and semiannual seasonality from 2009 to 2012. High relative risk areas and months should be the focus of greater attention in the prevention and control of HFMD. There was spatiotemporal interaction between HFMD relative risk. Adding a spatiotemporal interaction term could well explain the change of spatial effect with time, thus increasing the goodness of fit of the model. Meteorological factors, such as monthly average relative humidity, monthly average temperature, and monthly average rainfall, might be the driving factors of HFMD.

## Figures and Tables

**Figure 1 ijerph-16-01191-f001:**
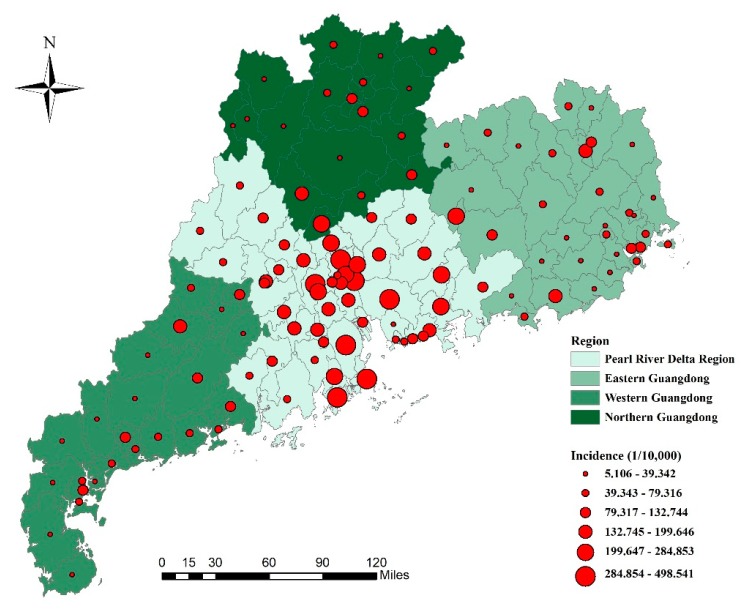
The four parts of Guangdong Province and cumulative incidence of hand, foot, and mouth disease (HFMD), 2009–2012.

**Figure 2 ijerph-16-01191-f002:**
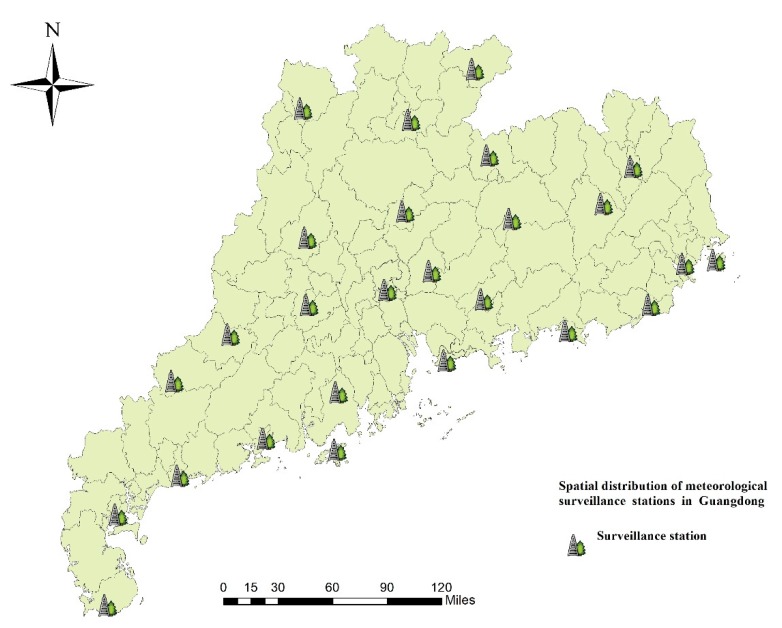
Spatial distribution of meteorological surveillance stations in Guangdong Province.

**Figure 3 ijerph-16-01191-f003:**
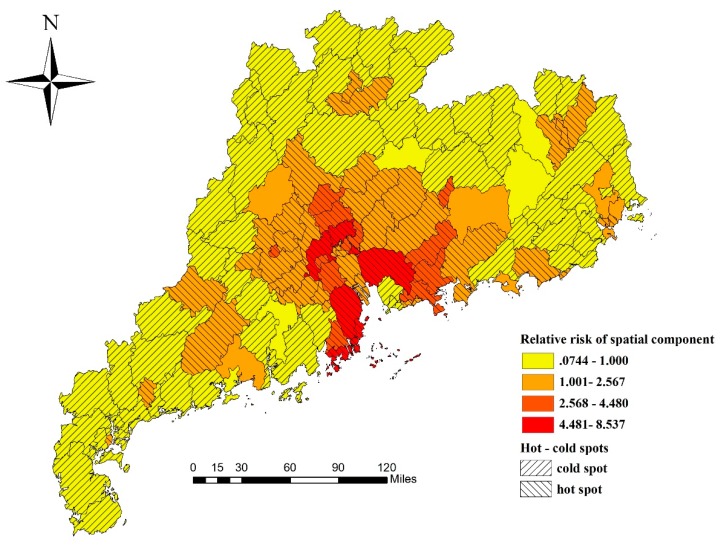
Relative risk (*RR*) values of the spatial effect and distribution of hot–cold spots.

**Figure 4 ijerph-16-01191-f004:**
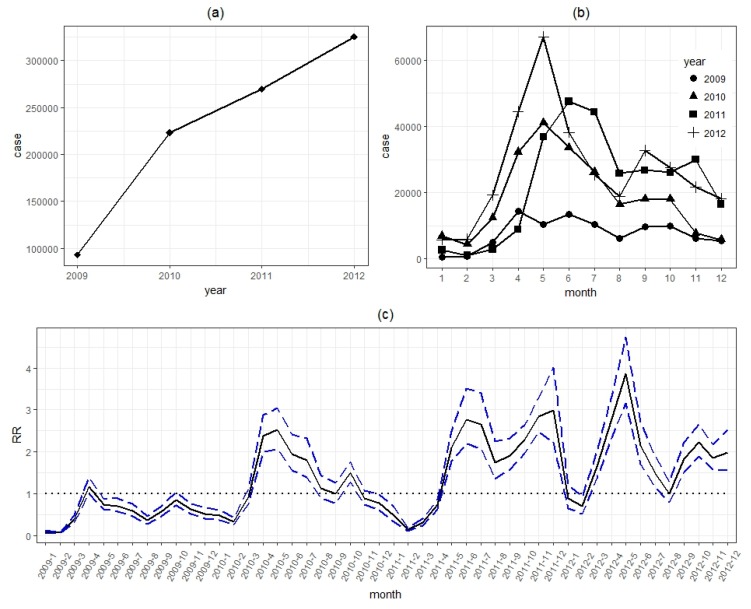
Annual number of cases of HFMD (**a**), monthly distribution of HFMD cases (**b**), and *RR* values of monthly effect (**c**) in Guangdong, 2009–2012.

**Figure 5 ijerph-16-01191-f005:**
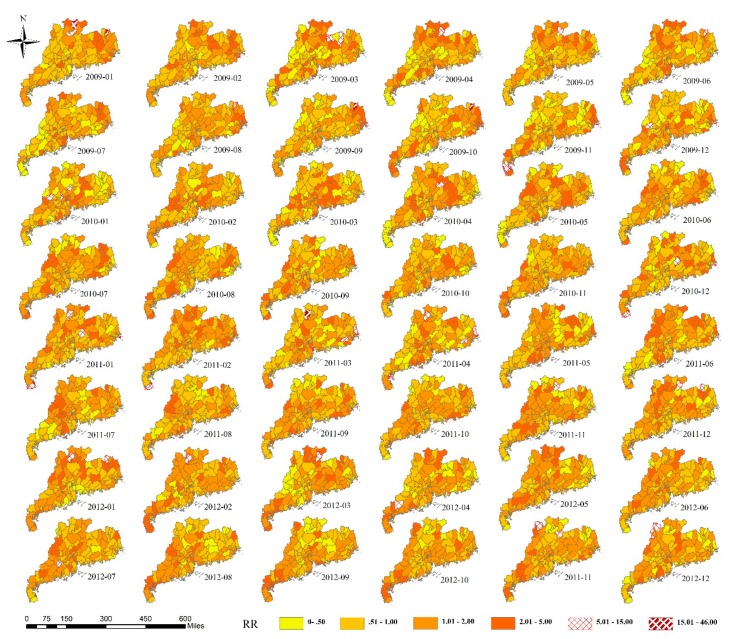
*RR* values of the spatiotemporal interaction effect.

**Figure 6 ijerph-16-01191-f006:**
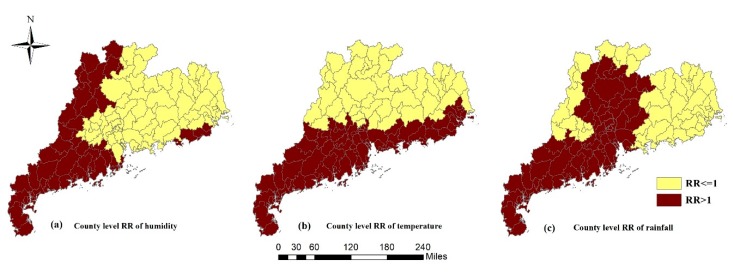
County-level estimated *RR* values of associated potential predictors.

**Table 1 ijerph-16-01191-t001:** Descriptive statistics of meteorological and socioeconomic variables.

Covariates	Minimum	2.5% Percentile	Median	97.5% Percentile	Maximum
Average relative humidity (%)	54.10	59.25	76.53	85.83	91.56
Average temperature (°C)	4.79	10.47	23.01	29.18	29.65
Average rainfall (mm)	0	1.66	98.76	410.11	645.50
Sunshine duration (hour)	18.91	38.05	149.18	253.48	303.47
Average wind speed (m/s)	1.25	1.53	2.10	3.38	5.33
Average GDP (yuan)	5717	9981	26,370	110,421	123,247

**Table 2 ijerph-16-01191-t002:** Multicollinearity evaluation results. VIF: variance inflation factor.

Covariates	VIF	Covariates	VIF
Average relative humidity (%)	2.305	Sunshine duration (hour)	3.126
Average temperature (°C)	3.827	Average wind speed (m/s)	1.091
Average rainfall (mm)	2.129	Average GDP (yuan)	1.025

**Table 3 ijerph-16-01191-t003:** The results of model selection. DIC: deviance information criterion.

	Model	Component	DIC
1	Non-spatiotemporal model	$\log(\theta_{ij})=b_{0}+\sum_{p}\beta_{p}X_{pij}$	1,083,638.0
2	Spatial model	$\log(\theta_{ij})=b_{0}+\sum_{p}\beta_{p}X_{pij}+\mu_{i}+\upsilon_{i}$	422,115.3
3	Temporal model	$\log(\theta_{ij})=b_{0}+\sum_{p}\beta_{p}X_{pij}+\gamma_{j}+\varphi_{j}$	816,474.6
4	Spatiotemporal model	$\log(\theta_{ij})=b_{0}+\sum_{p}\beta_{p}X_{pij}+\mu_{i}+\upsilon_{i}+\gamma_{j}+\varphi_{j}$	202,885.9
5	Spatiotemporal interaction model	$\log(\theta_{ij})=b_{0}+\sum_{p}\beta_{p}X_{pij}+\mu_{i}+\upsilon_{i}+\gamma_{j}+\varphi_{j}+\delta_{ij}$	45,036.5

**Table 4 ijerph-16-01191-t004:** Heterogeneity detection with *q*-statistic.

Data	*q*	*p*	Data	*q*	*p*	Data	*q*	*p*	Data	*q*	*p*
2009-01	0.122	0.856	2010-01	0.179	0.663	2011-01	0.145	0.771	2012-01	0.321	0.013 *
2009-02	0.292	0.095	2010-02	0.296	0.221	2011-02	0.238	0.302	2011-02	0.191	0.507
2009-03	0.336	0.022 *	2010-03	0.366	0.058	2011-03	0.089	0.980	2011-03	0.409	<0.001 *
2009-04	0.301	0.038 *	2010-04	0.251	0.184	2011-04	0.215	0.534	2011-04	0.379	0.004 *
2009-05	0.229	0.257	2010-05	0.190	0.507	2011-05	0.292	0.207	2011-05	0.465	<0.001 *
2009-06	0.265	0.362	2010-06	0.315	0.038 *	2011-06	0.274	0.112	2011-06	0.341	0.019 *
2009-07	0.287	0.202	2010-07	0.403	0.002 *	2011-07	0.498	<0.001 *	2011-07	0.325	0.033 *
2009-08	0.402	0.013 *	2010-08	0.458	<0.001 *	2011-08	0.299	0.069	2011-08	0.319	0.037 *
2009-09	0.254	0.198	2010-09	0.221	0.541	2011-09	0.260	0.203	2011-09	0.573	<0.001 *
2009-10	0.246	0.243	2010-10	0.281	0.265	2011-10	0.259	0.267	2011-10	0.290	0.101
2009-11	0.266	0.264	2010-11	0.179	0.710	2011-11	0.153	0.763	2011-11	0.158	0.715
2009-12	0.129	0.916	2010-12	0.174	0.589	2011-12	0.239	0.149	2011-12	0.264	0.096

* Indicated spatial heterogeneity.

**Table 5 ijerph-16-01191-t005:** Posterior means and confidence intervals of potential predictors.

Covariates	Unstandardized Coefficients (95% *CI*)	Standardized Coefficients (95% *CI*)
Intercept	0.054 (0.021, 0.138)	0.397 (0.359, 0.439)
Average relative humidity (%)	1.015 (1.006, 1.024)	1.102 (1.042, 1.166)
Average temperature (°C)	1.045 (1.021, 1.069)	1.291 (1.128, 1.476)
Average rainfall (cm)	1.004 (1.001, 1.008)	1.050 (1.008, 1.094)
Sunshine duration (10 h)	0.997 (0.985, 1.009)	0.982 (0.915, 1.054)
Average wind speed (dm/s)	0.998 (0.990, 1.007)	0.991 (0.951, 1.032)
Average GDP (1000 yuan)	0.999 (0.994, 1.003)	0.961 (0.835, 1.105)

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
