# Peer review of "Spatiotemporal Distribution of Hand, Foot, and Mouth Disease in Guangdong Province, China and Potential Predictors, 2009–2012"

_ijerph, 2019, doi:10.3390/ijerph16071191_

Round 1

Reviewer 1 Report

This paper is concerned with performing a spatiotemporal Distribution of Hand Foot and Mouth Disease in Guangdong Province, China and Potential Predictors. This study aimed to identify the spatiotemporal distribution characteristics and potential predictors for HFMD. I found it needs improvements and clarifications, and I would suggest that authors clarify these doubts the article.

Specific comments

1 What about the possible collinearity issues in the covariates, most importantly regarding the meteorological variables?

2 Do you consider other risk factors influence the occurrence of this disease? Why do you select just the few factors? what is the relative importance of these variables on HFMD?

3 How many weather stations in each locations? Would you please present the spatial distribution of these stations? How do you collect and spatial interpolate for climatic data?

4 Do you consider the confounders?      

5 There are some parts of the manuscript that are still confusing as regards the methodology used and its implementation step-by-step.

6 There are some other spatial-temporal model, such as spatial panel model, et al. Can the author(s) elaborate on their methodology selection?

7 The discussion and conclusion is one of the most interesting parts of the paper. The authors should highlight better their new contributions of their analysis as compared to the previous literature.

8 Line 159,269, “The spatial distribution of HFMD risk also exhibited explicit spatial heterogeneity”. The spatial heterogeneity should be quantified by statistic index, such as q statistics.

Ref: Wang JF, et al. 2016. A measure of spatial stratified heterogeneity. Ecological Indicators 67(2016): 250-256.

9 Line 116 , “We assumed that meteorological data did not vary significantly across years”, can he author(s) verify the assumption?

10 Line 190,”  In previous studies, spatial effects were assumed to be fixed and did not change over time. Add references.”, add references.

11 Table 2, what’s mean of “Range” and “Coefficients”?

12 Figure 6, what’s mean of “RR values of associated potential predictors.”? What are the exposure and control index?

Author Response

Thank you very much for giving us an opportunity to revise our manuscript. We appreciate the editor and reviewers very much for their constructive comments and suggestions on our manuscript. We have studied the comments carefully and made corrections. Hope meet with approval.

Reviewer 2 Report

The estimation of spatio-temporal heterogeneity of a contagious disease is a difficult and ambitious task. I commend the authors for that. However, the authors should try to address the following issues in their work.

1.       Weather data of 2009 is not representative of 2009-2012 period. See for example,

·         https://www.worldweatheronline.com/lang/en-au/guangzhou-weather-averages/guangdong/cn.aspx

·         https://ggweather.com/enso/oni.htm

·         It is possible this has played a role in such low RR for the environmental factors.

2.       Model is potentially ill-posed with identifiability and singularity problems.

·         Minor point- This is an empirical model based on assumptions and is not a truth of nature. As such all terms, “followed” should be replaced by “assumed to follow” or “assumed”.

·         Provide a justification for CAR model (or why not a SAR model), for example. Did your exploratory analysis suggest it? The justification could simply be a reference relevant for infectious diseases spatial epidemiology.

·         Figures~4 and 5 caption needs clarity. It is not immediately obvious what is the variable that has been depicted in Figure 5. All we see is possible heterogeneity in space-time. Heterogeneity is not-necessarily the same as spatio-temporal interaction. I suggest that the authors argue with example from the plots wht they mean by spatio-temporal interaction, here.

·         There are several grammatical flaws.

Major issues

·         The authors want to separately model the year effect and month effect (seasonality). For this they have suggested an additive combination of month and year effect- assuming orthogonality between these two effects. This however ignores the interaction between months and year. This is important because months are a subset of years. The correct framework for such modelling is SARIMA models that allows for serial and seasonal correlations. The data should have only one unit of time, either month or years, usually the smaller of the two units. Else singularity – via multicollinearity (for example) - is a common issue. The orthogonality assumption might still be considered if the authors had used monthly weather patterns for all months included in the study (and not just 2009).

·         Is there a possibility of identifiability between v_i and mu_i? A CAR model already includes an independent Gaussian component!

·         What is the rationale for selecting random walks for month and year once the authors have already modelled month and year effect?  I hope that the authors have considered the fact that including a random walk introduces an infinite variance non-stationarity on the process! Have the authors considered the implications of this in the context of HFMD prevalence/incidence? If the residual effects in your empirical model is a random walk it would often indicate that the proposed model is ill-posed. There are tests (unit root) that can be performed to assess if you really need a random walk assumption for a particular variable.

·         How does the estimation procedure differentiate (identifiability) between the additive components, γ and φ? The same goes for τ and ω.

·         Could you please mention the total number of parameters and what was the total sample size?  

·         What are the spatial and temporal resolution of all explanatory factors? That is, do you have all variables available at county level? If not please discuss the limitations of your model how are the data aggregated. Biases caused due to data aggregation are well studied. Please refer to relevant articles.

·         Deviance information criterion of an ill posed model is of little value.

3.       The study lacks novelty and reads like a replication study- in line with other provinces within China.

·         Also this reviewer could not be sure if proposing such a complex model has enhanced our understanding of the epidemiology of the disease given the numerous epidemiology studies on HFMD already reported from China (and the neighbourhood) on the same disease from other provinces.

·         Given the low risk ratios (RR) for the environmental factors one might produce the same maps simply by interpolating a monthly model of counts based at district level. That is, on the one hand the model seems ill-posed and on the other it would have questionable predictive impact.

4.      In summary the Introduction starts the well on the epidemiology side. But then ends with this statement- “Traditional methods, such as regression, time series analysis, spatiotemporal scan statistics, etc., 79 cannot be used to address the aforementioned knowledge gaps. Therefore, this study chose the 80 Bayesian spatiotemporal model, which added the spatiotemporal effect to the generalized additive 81 model, to identify spatiotemporal variations and the effects of potential predictors [20].” It is an unsubstantiated statement and does not follow scientific rigour.

Author Response

(The authors gave the same response as above.)

Round 2

Reviewer 1 Report

The manuscript has been significantly improved and now warrants publication in IJERPH.

Author Response

Reviewer's comments are highly appreciated. We have our manuscript professionally edited this time. Hope meet with approval.

Reviewer 2 Report

I have gone through the authors’ response. I commend the authors for considering my critique and for their detailed response, including many modifications and data related improvements in particular including rainfall data from 2009-2012.

However, as a statistician I stay circumspect about the choice of model complexity and (lack of) model validation.

 I also stay skeptical about the novelty of the epidemiology given the plethora of similar studies that already exists.

But the authors have worked hard and I would defer the decision to the Associate Editor.

Author Response

Reviewer's comments are highly appreciated, especially in statistical methods.

Regarding the model selection, compared with the generalized additive model (model 1) that without spatiotemporal effects, the model considering the spatiotemporal effects (model 5) significantly improved the goodness of fit. This is the advantage of the Bayesian spatiotemporal model used in this paper. However, we only used the DIC values to validate the model in the study; the stability of the model should be further explored in the future research.

In terms of novelty, this is the first Bayesian spatiotemporal model research in Guangdong Province. The model incorporated the spatiotemporal interaction effect, while this has not been considered in the previous studies. This is a comprehensive study of HFMD in terms of spatiotemporal effects and meteorological variables.

In addition, we have our manuscript professionally edited this time. Hope meet with approval.